# Supporting and Enhancing NICU Sensory Experiences (SENSE), 2nd Edition: An Update on Developmentally Appropriate Interventions for Preterm Infants

**DOI:** 10.3390/children10060961

**Published:** 2023-05-28

**Authors:** Roberta Pineda, Polly Kellner, Carolyn Ibrahim, Joan Smith

**Affiliations:** 1Chan Division of Occupational Science and Occupational Therapy, University of Southern California, Los Angeles, CA 90089, USA; 2Department of Pediatrics, Keck School of Medicine, Los Angeles, CA 90089, USA; 3Program in Occupational Therapy, Washington University School of Medicine, St. Louis, MO 63110, USA; 4Department of Health Sciences, Rush University, Chicago, IL 60612, USA; 5Department of Quality, Safety, and Practice Excellence, St. Louis Children’s Hospital, St. Louis, MO 63110, USA

**Keywords:** sensory-based interventions, sensory integration, sensation, exposure, environment, preterm, neonatal intensive care unit, NICU, tactile, auditory, multimodal, multisensory, vestibular, kinesthetic, visual, olfactory, gustatory, parenting, SENSE, review, program development

## Abstract

The Supporting and Enhancing NICU Sensory Experiences (SENSE) program promotes consistent, age-appropriate, responsive, and evidence-based positive sensory exposures for preterm infants each day of NICU hospitalization to optimize infant and parent outcomes. The initial development included an integrative review, stakeholder input (NICU parents and healthcare professionals), and feasibility focus groups. To keep the program updated and evidence-based, a review of the recent evidence and engagement with an advisory team will occur every 5 years to inform changes to the SENSE program. Prior to the launch of the 2nd edition of the SENSE program in 2022, information from a new integrative review of 57 articles, clinician feedback, and a survey identifying the barriers and facilitators to the SENSE program’s implementation in a real-world context were combined to inform initial changes. Subsequently, 27 stakeholders (neonatologists, nurse practitioners, clinical nurse specialists, bedside nurses, occupational therapists, physical therapists, speech-language pathologists, and parents) carefully considered the suggested changes, and refinements were made until near consensus was achieved. While the 2nd edition is largely the same as the original SENSE program, the refinements include the following: more inclusive language, clarification on recommended minimum doses, adaptations to allow for variability in how hospitals achieve different levels of light, the addition of visual tracking in the visual domain, and the addition of position changes in the kinesthetic domain.

## 1. Introduction

The Supporting and Enhancing NICU Sensory Experiences (SENSE) program was developed following research that identified alterations in the brain structure and function in hospitalized infants in low-stimulation NICU environments [1]. The SENSE program was designed to promote daily, positive, and evidence-based sensory experiences for premature infants who spend their first several months hospitalized in the NICU [2], which coincides with a period of rapid brain development [3,4,5]. The SENSE program considers the developmental trajectory of the preterm infant in the NICU and gives options for evidence-based sensory exposures and the recommended amount, based on the infant’s postmenstrual age (PMA). The program can be adapted to each specific infant’s needs and medical conditions.

The SENSE program was developed using a stepwise, rigorous, and scientific process, which included a review of the published literature from the previous 20 years to identify appropriate, evidence-based multisensory interventions for preterm infants in the NICU, along with the timing of when those interventions took place within each study [6]. Expert guidance was obtained from NICU healthcare professionals on the current use of sensory interventions [7], and parent input was sought to identify perceptions about having a guide on different evidence-based methods for parents to interact with their baby [8]. Focus groups at the proposed study site for research on the SENSE program aided an understanding of percieved feasibility and implementation. The process of the SENSE program’s development was outlined in a previous publication [9], and the SENSE program materials were made available to other hospitals upon request in 2018 to represent the current best practice related to sensory exposures in the NICU.

The SENSE program was also designed to encourage parents to be the providers of positive sensory exposures to their infants. Parents often struggle with role alteration when their young infants are hospitalized in the NICU [10], and many have challenges understanding what their role is in the NICU. The SENSE program provides parents with easy-to-understand activities that can be implemented on a daily basis with specific dose targets. Parents are asked to perform these activities with their infants each day of hospitalization to optimize their infant’s development, while also fostering their own presence and engagement in the NICU. Being given a concrete role and expectations has been shown to be associated with increased parent involvement and presence in the NICU [11,12,13,14]. Moreover, parent presence and involvement in the NICU is related to improved infant outcomes [15,16]. Therefore, the SENSE program was designed to optimize both infant and parent outcomes by putting the family at the center.

Since the launch of the initial SENSE program, research has been conducted to understand the impact of the program on infant and parent outcomes. A pilot study identified that, among families who received the SENSE program, mothers demonstrated improved confidence and infants demonstrated improved neurobehavior at the term-equivalent age [17]. Subsequently, a randomized clinical trial determined that infants who received the SENSE program had more lethargy on the NICU Network Neurobehavioral Scale [18]. The meaning of this finding remains unclear but may relate to increased relaxation and sleep among babies who receive nurturing interventions, which may be consistent with other research findings of improved sleep among infants experiencing kangaroo care [19]. Infants who received the SENSE program also had higher communication scores on the Ages and Stages Questionnaire at 1-year corrected age, although this relationship failed to reach significance after controlling for medical and social risks [18]. Another study found normal neurodevelopmental scores on the Hammersmith Neonatal Neurological Examination at the term-equivalent age, low parent stress, and high parent satisfaction following the implementation of the SENSE program [20]. The SENSE program has also been related to increased caregiver satisfaction of infant care in the NICU and improved feeding outcomes [21].

Research on the implementation of the program has identified that the SENSE program has good reach (was received by the targeted population), is effective and acceptable with minimal costs, and has good fidelity [22]. Another study on the feasibility of the SENSE program identified that recruitment and retention rates for preterm infants born between 28 and 33 weeks in a level III NICU were 87.5% and 100%, respectively [20]. Earlier parent education, within the first few days after preterm birth, has been related to more parent participation in SENSE program interventions throughout hospitalization [22], highlighting the powerful influence of early education that empowers parents to understand their important role and engage in the NICU. Further, the SENSE program has been related to increased parental presence and engagement among high-risk groups, such as families with younger mothers and parents living farther distances from the hospital [23].

There has been a large demand for the SENSE program among NICU clinicians worldwide since the program materials were made available to other hospitals in 2018. The cost of the program supports only the cost of its distribution, with no direct financial benefit to any person or entity.

Due to continuously emerging research on sensory exposures related to improved outcomes, we have committed to a new review of the evidence and engagement with an advisory team to inform changes to the SENSE program every 5 years. The purpose of this manuscript is to define the process for informing revisions, outline the decision-making process as to which changes should be made, and identify the refinements made to the SENSE program in 2022.

## 2. Methods

The original process of the SENSE program’s development has been previously described [9]. A comparable version of the original systematic, rigorous process to define the SENSE program was used to define changes for the 2nd edition. The update process consisted of a new integrative review to identify studies that had emerged within the past 5 years (since the time of the original review), a compilation of clinician guidance over the previous 5-year period, a survey of healthcare professionals who had obtained the SENSE program to define the barriers and facilitators to implementation in a real-world context, and engagement with a SENSE advisory team (consisting of a geographically diverse and multidisciplinary group of healthcare professionals and parents) to inform changes. We received IRB approval (USC IRB UP-20-01299, approved 1 April 2021) to conduct the survey, which is reported in another manuscript [24].

### 2.1. The Integrative Review

An updated integrative review was conducted through a systematic search for studies published from October 2015 to October 2020 using databases including MEDLINE (via PubMed), CINAHL (Cumulative Index to Nursing and Allied Health Literature), the Cochrane Library, and Google Scholar. Studies were included if they imposed a quantifiable environmental sensory exposure during the NICU stay, prior to 36 weeks PMA. The relevant outcomes included infant behavioral outcomes, neurobehavioral outcomes, parent well-being, and other parental outcomes. Studies with outcomes related solely to pain or feeding were excluded. Samples of healthy infants and studies with a sample size of <30 and without an a priori calculation of power were excluded. Two reviewers set the search engine and screened articles for inclusion. One reviewer performed data extraction that was checked for accuracy by a second reviewer. An assessment of study quality was independently performed by two reviewers using the United Kingdom’s National Institute for Health and Care Excellence (NICE). Study findings were summarized qualitatively, with evidence related to each type of sensory intervention defined across PMA. Fifty-seven new articles on sensory-based interventions (auditory, vestibular, visual, kinesthetic, and olfactory/gustatory) were identified that were used to improve outcomes for preterm infants and their parents in the NICU [25].

### 2.2. Clinician Guidance

Clinician input across the previous 5-year period (via email correspondence, social media dialogue, phone conversations and Zoom calls, as well as through direct conversation) was compiled. With the increased focus on small baby programs, much of the clinician guidance focused on an interest in further differentiating how to safely tailor SENSE interventions to the small baby.

### 2.3. Implementation Survey of Healthcare Professionals

The SENSE Program Implementation Survey was sent out to 211 hospitals who had obtained the SENSE program prior to 11 March 2020, to probe perceptions of the program and better understand implementation, costs, and adaptations made across sites. Among 114 respondents (54% response rate), important insights about acceptability, adoptability, barriers, and facilitators were identified, which informed changes to the parent education materials and administration and implementation manual. Information that guided modifications included that 33% of respondents indicated that they made adaptations to the SENSE program to facilitate implementation, and 64% of them were planned/proactive, while 36% were reactive. Most (64%) of these adaptations were determined by the therapist and had to do with adapting to the NICU context (58%). Some respondents indicated interventions in the SENSE program that their NICU did not endorse, including music (19%), massage (15%), talking to the infant during sleep (4%), use of vestibular stimulation (4%), and any interventions for infants < 26 weeks PMA (4%). There were 88% of respondents who indicated that they use the SENSE parent education materials, with most of them (80%) using the printed materials. There were 80% of respondents who indicated that they assess the tolerance of the infant, with 57% of them using the materials in the SENSE program to do so.

### 2.4. The SENSE Advisory Team

Once the new evidence, clinician guidance, and survey had been completed and results carefully considered, a SENSE advisory team (consisting of 3 neonatologists, 2 nurse practitioners, 1 registered nurse, 1 clinical nurse specialist, 1 neonatal nurse practitioner, 6 occupational therapists, 5 physical therapists, 3 speech-language pathologists, 1 music therapist, and 3 parents) was assembled. All healthcare professionals had a minimum of 5 years of clinical experience working in a NICU. The SENSE advisory team had the goal of ensuring that the SENSE program was evidence-based, incorporated the most current published literature, remained appropriate for different types of families, and was applicable to a variety of NICU settings. A call for members of the SENSE advisory team was made through email and social media groups for those interested in neonatal therapy research and those who were familiar with the original SENSE program. Each member was asked to commit 10 h for a period of up to one year to engage in online meetings, discussions, and individual review of materials. The larger 27-member group was broken into work groups of 2–7 individuals to weigh in on specific components of the program (such as tactile, auditory, visual, olfactory, kinesthetic, parent messaging, and other sections of the parent education materials). The work groups reviewed the new evidence and the changes suggested to the program. An iterative and interactive process occurred with the research team until near consensus was achieved on the suggested refinements to each section.

## 3. Results

All of the findings from the previous systematic and scientific process undertaken to develop the SENSE program, the new integrative review, clinician guidance, the implementation survey, and advisory team feedback were used to refine the SENSE program, which included improvements to the parent education materials and the implementation and administration manual.

The new integrative review (2015–2020) identified many interventions that had already been defined in the previous integrative review (1995–2015). However, the new integrative review provided evidence to support some additions, as well as modifications, to the timing and wording of the information contained in the SENSE program. Below, we identify the changes and rationale for the recommendations related to each sensory system, as well as for content in the Appendix A of the SENSE program.

### 3.1. Tactile

The previous integrative review identified evidence-based tactile interventions to be gentle human touch, skin-to-skin care, holding, and massage. The new integrative review expanded the evidence base for skin-to-skin holding to include its use as early as in the delivery room [26,27]. Therefore, the 2nd edition of the SENSE program continues to propose tactile exposure as early as ≤23 weeks PMA, with choices of skin-to-skin contact and hand hugs for a minimum of 1 h. The repertoire of tactile exposures expands across PMA to include the addition of holding for short periods as the infant starts to regulate their temperature and has the addition of massage starting at 32 weeks PMA. At term-equivalent age, the SENSE program provides choices of tactile exposures that include skin-to-skin care, holding, massage, or hand hugs for a minimum of 3 h. Additionally, due to new evidence supporting multimodal sensory experiences for preterm infants in skin-to-skin care, clarification was added to the parent education materials that the parent can add voice, music, or other tactile activities while doing skin-to-skin care, to the tolerance of the baby.

The timing for the introduction of massage received increased attention by the SENSE advisory team following the second integrative review. There is some emerging evidence supporting massage as early as 28 weeks PMA [28,29]. However, the use of massage this early is not clinically well established, and it was felt that infants between 28 and 32 weeks PMA have less capacity to tolerate the sensory input of massage and could experience undue stress. In addition, it was noted that the vast majority of the literature with evidence supporting massage includes massage that is performed after 32 weeks PMA [30,31,32,33]. Further, it was considered important to maintain a more conservative recommendation for massage due to the intention for parents to be the ones conducting it. Parents often do not have extensive training in how to carry out massage while reading and responding to cues for immature and fragile infants. For these reasons, starting at 32 weeks PMA is when massage is identified as an option for tactile exposures in the 2nd edition of the SENSE program, and it is the responsibility of the clinicians administering the SENSE program to determine what starting point for massage is appropriate for an individual infant. Despite the evidence that supports the use of massage, it was recognized that some NICUs do not endorse the use of massage as a tactile modality, while some cultures heavily focus on and find meaning and value in the use of massage.

The terminology for “gentle human touch” was changed to “hand hugs” in the 2nd edition of the SENSE program, as this terminology has grown in acceptance and was considered to be more parent-friendly than other terms such as facilitated tuck, containment, and gentle human touch.

### 3.2. Auditory

The previous integrative review identified evidence-based auditory interventions to be live music or singing, recorded sound (maternal voice, singing, and music), and recorded biological sounds. The new integrative review did not add additional interventions but did bring up some needed clarifications, such as the use of different decibel levels, use of headphones (versus not using headphones), type of music (such as pentatonic harp), and type of lullaby (such as Brahms lullaby). The auditory guideline remained the same in the 2nd edition of the SENSE program, with the use of quiet conversations at the bedside at the earliest PMAs, introducing reading and singing at the bedside at 28 weeks PMA, introducing music at 32 weeks if the infant tolerates it, and building up to a minimum of 3 h of intentional auditory exposure per day by term-equivalent age with a focus on an environment rich in language exposure.

There were studies from the new integrative review using recorded maternal voice and music prior to 32 weeks. One study using Brahms lullaby between approximately 29 and 34 weeks PMA identified differences in oxygen saturation levels, which was not a primary outcome of interest for our integrative review [34]. Another study using a recorded lullaby starting on the third day of life (in infants born 28–34 weeks) found no significant differences in physiology [35]. However, one study received increased attention by the research team and SENSE advisory team due to its use of recorded maternal voice estimated to have started as early as 25 weeks PMA, with the outcomes of this study related to development [36]. Following this review of the literature, there was much discussion about the use of natural (voice) exposures that are used in real-time to allow reciprocity and responding to infant cues in real-time. While the use of recorded sounds earlier than 32 weeks PMA has been used, specifically the parent’s recorded voice, it was felt that a live voice with reciprocity is preferred over the use of recorded sounds until the infant reaches 32 weeks, at which time the infant has better tolerance of sensory stimuli and there is more evidence for the use of music [37].

Decibel levels have also been a focus of attention. While the American Academy of Pediatrics and the World Health Organization recommendation for sound exposure is an average of 45 decibels, few NICUs meet this requirement [38,39]. The majority of reports suggest that decibel levels in the NICU average between 65 and 70 decibels [40,41], and there is some concern that sound exposures below this may not be perceived by the high-risk infant. However, just because sound exposures are high in the NICU does not justify an intervention to exceed the level it was intended to be at. Therefore, the recommendation in the 2nd edition of the SENSE program remains at a suggested average sound level of 45 decibels, and NICUs can continue to work toward decreasing other, non-natural, and/or noxious sounds. This is exceptionally important for sounds that exceed 68 decibels, as they have been identified as having a negative impact on physiology [42,43]. In addition, for the refinements to the SENSE program, it was felt that exact decibel levels were not in parent-friendly language, so 45 decibels as the target for auditory exposures was removed from the parent education materials and replaced with qualifying the volume of quiet conversations at the bedside as the “sound of a whisper”. The implementation and administration manual, however, identifies exact decibel levels to aid healthcare professionals in a quantifiable sound level to target.

### 3.3. Olfactory

The previous integrative review identified maternal scent and breast milk as the only evidence-based olfactory exposures. Clarifications about the use of breast milk scent were added to the 2nd edition, as the original program largely focused on the smell of the parent. A discussion occurred regarding the addition of an assigned time recommendation for olfactory exposure, as it is one of the only senses in the program that did not have a time target. This idea of a time target also heightened discussions about the use of a constant smell and whether infants might habituate to it. In addition, there was discussion about the potential impact and associated learning if a scent was placed by an infant and then they had multiple noxious stimuli that occurred. Based on evidence supporting the use of the scent of the parent and breast milk for 3 h [44], this minimum target was added to the 2nd edition for olfactory exposure across PMA. Additionally, discussion occurred regarding different ways for a parent to provide their scent to their infant. Scent cloths continue to be recommended, as well as providing clothing or bedding for the infant with a parent’s scent when parents are unable to visit. Clarification of how close parent contact provides parent scent was made, including how parent scent via close parent contact is the preferred method of providing olfactory stimuli.

### 3.4. Visual

In the previous integrative review, interventions largely focused on the light environment, specifically on the use of cycled light, starting at 32 weeks PMA. There was also some evidence on the use of the parent’s face for visual exposure closer to term-equivalent age. The new integrative review identified positive outcomes associated with having infants fix and follow a toy or human face starting at 34 weeks PMA [29]. Therefore, the 2nd edition encourages face-to-face interaction with a parent starting at 34 weeks PMA. Because the SENSE program aims to put parents at the center of all interventions, we have recommended that a human face be the stimulus being focused on and followed, rather than a toy.

There is not strong evidence to support cycled light prior to 32 weeks [45,46], which is why the SENSE program identifies cycled light at 32 weeks PMA. However, there is insufficient evidence to say that cycled lighting should not take place prior to 32 weeks PMA. Advisory team input, and the results of the latest integrative review, informed a change from advising “lights off” prior to 32 weeks PMA to recommending that the infant be protected from direct or bright light during this period. This is because there is insufficient evidence to indicate that a continuously dark environment at any time period is appropriate. Therefore, the 2nd edition of the SENSE program has more options for the light environment to have some flexibility of what is performed prior to the cycling of light at 32 weeks PMA, but includes the recommendation to avoid direct and bright lights throughout hospitalization.

For cycled light, a specific time (i.e., 12 h on and 12 h off) has been replaced with “during the day” and “during the night” in the 2nd edition to allow NICU staff to find a pattern that works for them (when the sun goes up/down, when nursing shift changes, etc.). Another important factor discussed was the benefit of gradual light transitions as in normal day/night rhythms. This is important, because sudden changes in light can impact sleep [46,47,48]. Slow transitions between day and night cycles can be achieved with careful attention to light levels during the transition or through the use of a dimmer switch. In units without dimmer switches, items such as curtains or blankets can be used over the incubator or windows to slowly introduce or decrease lighting. Such adaptations to the light environment were added to the implementation and administration manual.

Another important discussion point was light intensity. The literature identifying the impact of cycled light has variable lux levels ranging from 75 to 500 lux during the daytime [49], with most studies reporting levels around 250 lux [50,51]. Most studies indicate nighttime levels of <25 lux [51]. Every NICU is different, with some having significant natural light and others relying completely on artificial light. Achieving a specific target lux level will then be different across NICUs and even different among bedspaces within the same NICU. The current literature also indicates that a difference of 60–100 lux is needed for entrainment [51,52,53], so ensuring that a difference between day and night of at least that much is important. For the 2nd edition of the SENSE program, in keeping with flexible light levels, daytime light levels are described as “moderate office lighting”, with the hope for levels to be around 250 lux, and with darkness at nighttime preferred to be <25 lux. Implementation strategies and quantifiable light levels are discussed in the implementation and administration manual.

### 3.5. Kinesthetic

The previous integrative review included interventions described as physical therapy, but with outcomes largely focused on bone health. The new integrative review expanded the repertoire of evidence-based kinesthetic interventions to include guided movement and position changes. There is also emerging evidence to support passive range of motion (PROM), but since the positive outcomes in these studies were related only to bone health and weight gain (and not to neurobehavior), PROM was not added to the SENSE program at this time. The guideline continues to include opportunities for free movement and supporting the infant’s smooth movements when appropriate (i.e., when the infant demonstrates cramped synchronized movement). Clarifications were added that infants at low PMA may not tolerate free movement, and that such free movement can be contained and/or occur within the confines of positioning devices or by allowing movement of one extremity at a time, especially when performed with the 23–28 week PMA population.

The 2nd edition of the SENSE program also now includes recommendations that the infant experience at least two different positions each day for a minimum of 10 min each (e.g., side lying, prone, supine, upright sitting, and skin-to-skin) at the earliest PMAs. By 34 weeks PMA, the recommendation is for infants to experience being in at least three different positions each day, with tummy time being one of them. These positions may occur at any time throughout the course of a 24 h period and may already be encouraged based on position change recommendations that are part of standard care. Tummy time is important throughout infancy [54], and encouraging tummy time as a position in which infants should spend supervised time while in the NICU can lay the foundation for continued practice after discharge [55,56,57]. This time in prone is intended to be supervised with a parent present, so it aligns with safe sleep guidelines. The use of position changes throughout hospitalization is intended to provide important sensory and motor experiences for the infant. This is a different approach and context than position changes given to ensure pressure relief to maintain skin integrity.

### 3.6. Parent Education Materials

Parent education materials for the 2nd edition of the SENSE program consist of an 88-page booklet written at a 5th to 8th grade reading level that includes a glossary of common NICU terms along with chapters that detail the importance of parents and why their involvement matters, development of the senses, specific guidelines for how to provide sensory experiences, guidance on when to interact based on their baby’s cues, a “how to” guide with recommendations for sensory experiences to share with their baby each day of hospitalization with “doses” to target that evolve across PMA, and detailed instructions and videos for how to share sensory experiences (such as massage or gentle rocking) with their baby. This parent booklet is now available in English, Spanish, Arabic, French, Korean, Hindi, Chinese, and Hebrew, with additional translations ongoing. In addition to a print version of the booklet, these materials are also accessible electronically via a QR code to enable easy access on smart phones and/or tablets.

General changes were made to the wording in the 2nd edition of the SENSE program based on the results of the integrative review, advisory team feedback, survey responses, and informal clinician feedback. The chapters of the parent education booklet were re-organized and modified for inclusion, clarity, and promotion of parent engagement. Minor formatting changes were made to improve the aesthetics of the program and enhance the clarity of the content.

In recognition of evidence regarding gender-affirming care [58], the language surrounding parent and infant roles was degendered, and the language and terminology were expanded to be inclusive of different family structures. However, it was noted that some of the gender-neutral language in the 2nd edition could not be carried through to all of the translations, due to no direct translation being available. The language surrounding dosing of different sensory experiences was modified to emphasize that these recommendations are minimums, in an effort to encourage increased engagement with the interventions as appropriate based on regular assessment by the SENSE administrator. However, there was also a discussion about the use of language to ensure it does not induce added stress or feelings of guilt among parents. Therefore, wording was added acknowledging that parents may not be able to do everything in the book but to “do what they can”. It was also clarified that some infants may not be able to tolerate everything in the program. Some photos were updated, and additional pictures were identified as targets for future updating to be more inclusive of different ethnic groups and family structures.

### 3.7. Infant Assessment of Tolerance

The weekly infant assessment of tolerance is included as an optional component of the SENSE program to allow clinicians to formally evaluate whether infants are tolerating the interventions as described. In a large level IV NICU, it was found that 95% of infants tolerated the original SENSE program as described, but 5% required temporary adaptations based on a medical status that typically lasted a few days to a few weeks before returning to the program recommendations [22]. Experienced clinicians may choose to use alternate, more gestalt methods of determining infant tolerance rather than the formal infant assessment of tolerance included in the program. The infant assessment of tolerance largely stayed the same, but under medical interventions, a place to mark if the infant was in a minimal stimulation period (usually in the first 72 h) was added, as this could guide adaptations that are made to the program within this context.

### 3.8. Implementation and Administration Manual

The implementation and administration manual is a supplemental resource that serves as a “user manual” for clinicians to implement and support the program, with information regarding the development of the program, strategies for implementation, and adaptations that can be made to the program. Changes to the implementation and administration manual of the SENSE program were made based on the SENSE implementation survey, clinician guidance, and the advisory team’s feedback. A definition of the SENSE program was added, along with a rationale for the program and a detailed description of the process of developing the SENSE program. The ongoing commitment to evidence-based programming was emphasized, with the intention to update it every 5 years. The manual was reorganized to guide clinicians in three main subsections: components of the program, necessary personnel, and materials to assist in implementation. The added items to assist with implementation were a map of an implementation strategy, an implementation checklist, and examples of adaptations that can be made based on individual and organizational factors. It was also clarified that the process of implementation can be an iterative process of implementation, assessment of progress, and then adaptation. Specific lux levels and decibel levels to go along with the program recommendations were added to the manual for clinicians to have quantifiable targets, with suggested tools to aid clinicians in assessing light and sound in their NICUs. Additional factors surrounding the first 72 h were added to aid clinicians in evidence-based practice and determine choices of sensory exposures that align with their hospital’s policies and culture. The manual was also updated to include strategies for engaging families, provide applause for small progress toward positive sensory exposures, and outline recommendations to integrate the program into the context of daily life in the NICU.

## 4. Discussion

This manuscript discusses changes to the SENSE program that were made in 2022. The goal was, and will continue to be, to keep the SENSE program aligned with the latest evidence. The refinements made ensure that the program stays evidence-based and meets the needs of infants, families, and healthcare providers worldwide. The program will continue to be updated every 5 years using a similar process of an integrative review of new evidence, advisory team input, and survey feedback from hospitals that have obtained the program.

While the research to date has identified associations of the SENSE program with parent confidence, as well as infant neurobehavior and outcomes [17,18,21], more research is needed. Understanding its effect on high- and low-risk populations with different medical trajectories will aid our understanding of its impact. Further, understanding the impact of different sensory exposures and potentially parceling out the program to understand the unique contribution of each of the sensory exposures listed in the SENSE program could lead to refinements to decrease the program’s complexity and ease implementation in the future. Future research can also improve our understanding of implementation of the SENSE program across different levels of NICUs, in different geographic locations, and in NICUs with different policies, procedures, and team make-ups. While one study has now identified the benefits of the SENSE program being parent-led, rather than sensory exposures conducted by other healthcare professionals or volunteers [23], future research on the impact of parents engaging in SENSE program interventions throughout their infant’s hospitalization is warranted. Finally, the integration of the SENSE program with other programs in the NICU (such as Family-Integrated Care (FiCare), Newborn Individualized Developmental Care and Assessment Program (NIDCAP), Creating Opportunities for Parent Empowerment (COPE), Close Collaboration with Parents, and Family Nurture Intervention) deserves some attention, as hospitals strive to improve their support for early development and optimize long-term outcomes for all infants receiving life-sustaining medical care.

### 4.1. Limitations

While a systematic and rigorous process was used to refine the SENSE program, there were some limitations. The integrative reviews that now encompass 25 years of research were limited based on the inclusion and exclusion criteria, and many of the studies lacked clarity of when the interventions were conducted. This led to assumptions being made with the available information. The informal feedback from clinicians that was compiled over the previous 5 years was not formally analyzed using qualitative techniques, and it could be subject to bias based on receiving responses only from individuals who reached out. The implementation survey is limited by being a survey, rather than a direct observation of the SENSE program implementation. The timing of this survey was also challenging due to the impact of the COVID-19 pandemic on medical care in the NICU. The SENSE advisory team was made up of parents and healthcare professionals who were willing to volunteer their time to aid this process and may not reflect all the views of parents and healthcare professionals. When combining all the information through these processes, choices and decisions were not always clear, and discussions had to occur until near consensus was achieved. Despite the limitations, we have confidence that the 2nd edition of the SENSE program is evidence-based, incorporates stakeholder feedback, and is best practice in terms of sensory exposures for preterm infants in the NICU.

### 4.2. Impact

There has been a high demand for the SENSE program, especially in the United States, but also extending to other countries including Canada, Italy, Malta, Mexico, Chile, Switzerland, Spain, Taiwan, the United Arab Emirates, Qatar, Australia, South Africa, Argentina, New Zealand, Costa Rica, Turkey, the United Kingdom, Singapore, India, and Israel. More than 400 hospitals have obtained the program worldwide. Updates to the program will ensure that the thousands of infants and families who are reached by the program are receiving sensory experiences that have the power to improve hospital, family, and infant outcomes.

## Data Availability

Not applicable.

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
