# Peer review of "Supporting and Enhancing NICU Sensory Experiences (SENSE), 2nd Edition: An Update on Developmentally Appropriate Interventions for Preterm Infants"

_children, 2023, doi:10.3390/children10060961_

Round 1

Reviewer 1 Report

I am grateful for the opportunity offered to me to review this manuscript, which I find interesting in terms of its subject matter and current in terms of childhood cancer.

I would like to make some comments on it, in order to try to improve it and once it is published it will be of a higher scientific quality.

It is recommended that the introduction be enriched with a bibliography, so that it is more than 9 references.

It seems desirable in the evaluation of the studies, that this would have been tabulated with a score, in order to know whether, as they indicate later in the body of the article, the quality of the studies. This information under table is collected and contemplated to improve the reader's orientation.

The bibliography within the text should be placed according to the journal's rules. It should be in square brackets, but the full stop should be after the bracket and not before it.

I think there must be a correlation of the bibliography in the development of the article, so from number 16 it goes to number 18... and this happens with other bibliographies cited in the references section.

Likewise, I think the discussion should be enriched and a more profound discourse should be formalised.

Author Response

  1. It is recommended that the introduction be enriched with a bibliography, so that it is more than 9 references. The introduction contains 22 references. In order to keep it relevant and concise, no additional references have been added, as the current references embody the evidence to support each piece of content in the introduction.
  2. It seems desirable in the evaluation of the studies, that this would have been tabulated with a score, in order to know whether, as they indicate later in the body of the article, the quality of the studies. This information under table is collected and contemplated to improve the reader's orientation. The assessment of quality is reported in the publication on the integrative review, and as such it does not seem appropriate to include in detail in this particular manuscript. We have added to the manuscript that the assessment of study quality was independently performed by two reviewers using the United Kingdom’s National Institute for Health and Care Excellence (NICE). (Pineda R, Kellner P, Guth R, Gronemeyer A, Smith J. NICU sensory experiences associated with positive outcomes: an integrative review of evidence from 2015-2020 [published online ahead of print, 2023 Apr 7]. J Perinatol. 2023;10.1038/s41372-023-01655-y. doi:10.1038/s41372-023-01655-y).
  3. The bibliography within the text should be placed according to the journal's rules. It should be in square brackets, but the full stop should be after the bracket and not before it. This has been amended.
  4. I think there must be a correlation of the bibliography in the development of the article, so from number 16 it goes to number 18... and this happens with other bibliographies cited in the references section. This has been amended.
  5. Likewise, I think the discussion should be enriched and a more profound discourse should be formalised. This article is the second of its kind. Following the initial development of the SENSE program, such combination of the scientific and systematic process undertaken in its development was reported in Early Human Development.  Since the current article follows this same framework to inform changes to the program in the second edition, we have used the framework from the EHD  We also seek to not go beyond what the paper intends to do which is to define the process, what was done in response to the process, and what the final changes to the SENSE program were.

Reviewer 2 Report

Thank you for this important and thorough update on the SENSE II program. This work will be incredibly helpful to those already using SENSE and those who might use SENSE in the future. Revisions are requested to provide data to support the changes narrated in the results section of the paper.

·       Please note and cite what EQUATOR network guideline you used to guide manuscript writing, and consider using the checklist as a supplementary file

·       Please consider adding the survey questions you delivered as a supplementary file

·       I am surprised that results from the survey are not presented in tabular form – is this published elsewhere? Please provide demographics, etc. major results in a table

·       Please consider providing a table / supplemental file with the demographics/experience of the 27 members of the advisory panel. Or provide a citation if the panel has been used in other work

·       Given your rigorous methods, I am surprised that there isn’t quotes or documentation from your qualitative sources (emails, phone calls, Zoom calls, etc.). Please include a table with sample quotes to support your results. Or sample quotes supporting the changes you made within the results section would be helpful as well. While formal qualitative methods are lacking and is a limitation, there should at least be some documentation / data supporting the changes narrated in the results.

·       For olfactory, what is the evidence on oral immune therapy (mouth care with breastmilk) – could you clarify when you say breastmilk is an evidence based intervention (e.g., oral feedings, paci dips with breastmilk, and oral care)?

·       Could you clarify that “the infant spends at least 10 minutes in 2 different positions?” – do you mean back to back? Could/should this be implemented during cares?

Author Response

  1. Please note and cite what EQUATOR network guideline you used to guide manuscript writing, and consider using the checklist as a supplementary file: https://www.equator-network.org We have reviewed all the studies and checklists in the Equator guideline. This manuscript combines multiple studies into a process undertaken in order to update the SENSE program.  As such, none fit well. However, the AGREE reporting checklist fits the best.  We have included this checklist with the manuscript.
  2. Please consider adding the survey questions you delivered as a supplementary file. We have another article that has been submitted for publication detailing the survey results. Therefore, it would not be appropriate to include it here again. However, the hope is that once it is published, we can include it as a citation in the current article.
  3. I am surprised that results from the survey are not presented in tabular form – is this published elsewhere? Please provide demographics, etc. major results in a table. The results of the survey are included in a manuscript that has been submitted elsewhere. If that manuscript is accepted/published prior to this manuscript being accepted/published, we will cite accordingly.
  4. Please consider providing a table / supplemental file with the demographics/experience of the 27 members of the advisory panel. Or provide a citation if the panel has been used in other work. We have added professional designations, but a supplemental file with full credentials felt excessive for the purposes of this publication.
  5. Given your rigorous methods, I am surprised that there isn’t quotes or documentation from your qualitative sources (emails, phone calls, Zoom calls, etc.). Please include a table with sample quotes to support your results. Or sample quotes supporting the changes you made within the results section would be helpful as well. While formal qualitative methods are lacking and is a limitation, there should at least be some documentation / data supporting the changes narrated in the results. This is an interesting thought to include rigorous qualitative methods in the future, however, this was not employed as part of the study procedures for this SENSE program update. The advisory board members are not considered research subjects, and the iterative process allowed for professional exchange and the suggestions/changes to occur until near consensus achieved.
  6. For olfactory, what is the evidence on oral immune therapy (mouth care with breastmilk) – could you clarify when you say breastmilk is an evidence based intervention (e.g., oral feedings, paci dips with breastmilk, and oral care)? The SENSE program includes interventions with evidence to support positive developmental outcomes. The current literature on oral immune therapy/tastes of breast milk has not yet met inclusion criteria for our integrative review and thus not met inclusion criteria for SENSE program interventions. Another important distinction is “positive oral sensory experience” versus ‘oral care’, which are different constructs.
  7. Could you clarify that “the infant spends at least 10 minutes in 2 different positions?” – do you mean back to back? Could/should this be implemented during cares? The following sentence was added “These positions may occur at any time throughout the course of a 24 hour period and may already be encouraged based on position change recommendations that are part of care.”

Round 2

Reviewer 1 Report

The authors are thanked for their corrections and appreciate the higher quality of the research. The adjustment of the references and the inclusion of some of them justify the study and provide it with greater internal validity.

Reviewer 2 Report

Thank you for your responsive changes to reviewer comments. If the paper is meant to be a broad overview of Sense 2nd edition, then I believe the paper is appropriate to publish without additional changes. I was disappointed that the results of the survey will be reported elsewhere in a manuscript that has not been accepted for publication yet. While I do think the manuscript would still benefit from a tabular overview of changes, with supporting sample evidence, I understand if that is outside the scope of this paper (see below for previous review comment):

·         Given your rigorous methods, I am surprised that there isn’t quotes or documentation from your qualitative sources (emails, phone calls, Zoom calls, etc.). Please include a table with sample quotes to support your results. Or sample quotes supporting the changes you made within the results section would be helpful as well. While formal qualitative methods are lacking and is a limitation, there should at least be some documentation / data supporting the changes narrated in the results.